# Multiplex Label-Free Kinetic Characterization of Antibodies for Rapid Sensitive Cardiac Troponin I Detection Based on Functionalized Magnetic Nanotags

**DOI:** 10.3390/ijms23094474

**Published:** 2022-04-19

**Authors:** Alexey V. Orlov, Juri A. Malkerov, Denis O. Novichikhin, Sergey L. Znoyko, Petr I. Nikitin

**Affiliations:** 1Prokhorov General Physics Institute of the Russian Academy of Sciences, 38 Vavilov St, 119991 Moscow, Russia; jurimalkerov@yandex.ru (J.A.M.); nammen@yandex.ru (D.O.N.); znoykos@yandex.ru (S.L.Z.); 2National Research Nuclear University MEPhI (Moscow Engineering Physics Institute), 31 Kashirskoe Shosse, 115409 Moscow, Russia

**Keywords:** molecular biomarkers, detection of biomolecules, superparamagnetic iron oxide nanoparticles, high-throughput sensing, cardiac markers, real-time optical biosensors, kinetic rate constants, point-of-care, lateral flow magnetic immunoassay

## Abstract

Express and highly sensitive immunoassays for the quantitative registration of cardiac troponin I (cTnI) are in high demand for early point-of-care differential diagnosis of acute myocardial infarction. The selection of antibodies that feature rapid and tight binding with antigens is crucial for immunoassay rate and sensitivity. A method is presented for the selection of the most promising clones for advanced immunoassays via simultaneous characterization of interaction kinetics of different monoclonal antibodies (mAb) using a direct label-free method of multiplex spectral correlation interferometry. mAb-cTnI interactions were real-time registered on an epoxy-modified microarray glass sensor chip that did not require activation. The covalent immobilization of mAb microdots on its surface provided versatility, convenience, and virtually unlimited multiplexing potential. The kinetics of tracer antibody interaction with the “cTnI—capture antibody” complex was characterized. Algorithms are shown for excluding mutual competition of the tracer/capture antibodies and selecting the optimal pairs for different assay formats. Using the selected mAbs, a lateral flow assay was developed for rapid quantitative cTnI determination based on electronic detection of functionalized magnetic nanoparticles applied as labels (detection limit—0.08 ng/mL, dynamic range > 3 orders). The method can be extended to other molecular biomarkers for high-throughput screening of mAbs and rational development of immunoassays.

## 1. Introduction

Cardiac troponin I (cTnI) is a specific marker of acute myocardial infarction (AMI), which is one of the major mortality factors throughout the world [1,2,3,4]. Rapid and reliable distinction between AMI and other disorders is often critical for the patient’s life [1,5,6]. The clinical significance of cTnI was highlighted by a new generation of highly sensitive assays that enabled early differential diagnosis of AMI [7,8,9,10,11]. These assays, capable of measuring trace amounts of cTnI in healthy people, can detect a subtle rise of cTnI as early as within the first hours when cTnI concentration in blood is still extremely low [12,13,14].

There are a wide variety of highly sensitive assays for measuring cTnI in blood and serum [15,16,17,18]. However, those primarily implemented in stationary laboratories, such as the bead-based electrochemiluminescence immunoassay, meet the stringent requirements for cardiac clinical applications [19,20,21]. These requirements include ultrasensitivity and quantitative results to allow monitoring of the cTnI concentration dynamics [11,22]. Substantial efforts have been put to developing portable express cTnI tests that can be executed at a point of care (POC), e.g., in emergency at a pre-hospital stage without complicated equipment [20,23,24,25]. In this respect, the methods based on the lateral-flow principles are frontrunners [26,27,28]. The results are evaluated either visually or with portable optical analyzers. Some POC tests have already been used by ambulance services for express troponin detection [17,29,30]. However, these tests generally cannot compete in sensitivity with the laboratory-based techniques. Thus, the express methods for quantitative POC-cTnI registration with analytical characteristics, which are not inferior to those of the stationary laboratory techniques, are still to be developed.

One of the key factors that strongly affects sensitivity and rapidity of an immunoanalytical method is the selection of antibodies, which feature the best equilibrium and kinetic constants of association/dissociation to ensure rapid and specific formation of the antibody-antigen complex [31,32,33]. The kinetic characterization is performed using different label-based and label-free techniques [34,35,36,37,38,39]. Since a label may hinder or even inhibit the analyzed interaction, the label-free techniques represent a gold standard for the quantitative assessment of kinetic parameters of antibody-antigen interactions [40,41,42]. The surface plasmon resonance biosensors based on sensor chips with precisely deposited conductive films are most common. Despite recent substantial advances in the label-free technologies [43,44,45,46,47], the task of development of the biosensors, which use affordable consumables for characterization and selection of antibodies in high-throughput microarray format, is yet elusive.

In this research, for the mentioned task, we used an original label-free biosensor based on the multiplex spectral correlation interferometry (mSCI). We showed high-throughput kinetic characterization of different monoclonal antibodies (mAb) to cTnI. By means of standard microarray spotting on an epoxy-modified glass surface without any additional conductive layers, a pair of capture/tracer antibodies was selected that exhibited the most rapid and efficient recognition of antigen. The selected mAbs were used for the development of sensitive express lateral flow assay based on magnetic nanotags detectable with the method of magnetic particle quantification (MPQ). The MPQ reader compared favorably to other magnetic sensors as readers for LF test strips due to the record-breaking sensitivity of 60 zM [48] and unrivalled seven-order linear range of quantitation of magnetic nanotags irrespective to their color and depth inside the lateral flow membrane [49]. The developed MPQ-based cTnI assay showed superior analytical features: assay time—25 min, limit of cTnI detection in human serum—0.08 ng/mL, the dynamic range more than three orders of concentration magnitude.

## 2. Results and Discussion

### 2.1. Multiplex Spectral Correlation Interferometry

Simultaneous measurements of kinetics for several antibodies were implemented by an updated compact biosensor (see photo in Figure 1) based on multiplex spectral-correlation interferometry, the principle of which is described in detail in [41,50]. Briefly, in this method, a radiation from a superluminescent diode (λ = 830 nm, 20-nm spectral width at half-height) passes through a scanning Fabry–Perot interferometer (sFPI). A piezoelectric driver periodically changes the sFPI base (distance L between two sFPI mirrors). The radiation is then incident on a microscope cover slip that serves as both a sensor chip and a two-beam interferometer. The periodically modulated radiation intensity U(L) is recorded with peaks that appear when maximums in the sFPI transmission spectrum coincide with those in the reflection spectrum. Such a coincidence occurs when the sFPI base is equal to the optical thickness of the sensor chip with the biolayer (or differs by λ/2) in the analyzed point on the chip. Upon adsorption/desorption of biomolecules, the optical thickness of the sensor chip with the biolayer changes. The changes Δd in the optical thickness of the sensor chip with the biolayer are determined by measuring the phase shift of the U(L) function. The Δd value is averaged over the registration spot area, where the antigen is immobilized. The temporal dependence of Δd is recorded in real time throughout an experiment. The inert spots on the sensor chip are used as the reference channels to eliminate temperature drifts due to possible thermal expansion of the cover slip [51].

### 2.2. Microarray-Based Setup for Multiplex Label-Free Kinetic Characterization of Antibodies

The kinetic characteristics of different clones of monoclonal antibodies to cTnI were studied using the multiplex label-free method of spectral correlation interferometry (Figure 2). As single-used sensor chips, standard microscope cover glass slips with epoxy-silane surface modification were employed. Prior to the label-free sensing, several monoclonal antibodies were covalently immobilized in different recognition spots on the epoxylated glass surface of a single sensor chip. The used immobilization technique is convenient for label-free biosensors because the epoxylated glass sensor chips can be stored for a reasonably long time without quality impairment, providing strong covalent sorption without need of activation [52,53,54]. In the experiments, the sensor chip with the immobilized mAbs was placed into an mSCI-biosensor that quantitatively registered in real time variations in the thickness of a biolayer on the sensor chip surface, which occurred due to biochemical reactions. At first, a solution of cTnI antigen was pumped over the whole surface of the multi-spot sensor chip. The realized simultaneous and independent registration of the antigen binding in each registration spot enabled rapid and efficient characterization of all immobilized monoclonal antibodies. The dissociation of the formed immune complexes was registered at the second stage during pumping of the buffer without antigen. The recorded temporal dependences (sensorgrams) of the biolayer thickness were fitted by exponential functions to determine the kinetic constants of association and dissociation followed by calculation of the respective equilibrium constants. 

The proposed design of experimental setup offers virtually unlimited potential for multiplexing due to compatibility with a high-throughput microarray format on glass sensor chips. Accordingly, it enables simultaneous characterization of a multitude of different antibodies.

### 2.3. Measuring the Kinetic and Equilibrium Constants for Capture Antibodies

In this research, the proposed method of rapid and simultaneous characterization of monoclonal antibodies was applied to six different commercially available clones of antibodies to cardiac troponin I: 19C7, 16A11, 4C2, M155, MF4, and 8E10. Figure 3 shows the sensorgrams recorded for each clone by the mSCI-biosensor. These sensorgrams illustrate both stages of association and dissociation. The calculated values of kinetic and equilibrium constants of interaction of these clones with cTnI antigen are given in Table 1.

As one may see from Table 1, the minimal values of equilibrium constant of dissociation for 16A11, 19C7, and 4C2 clones indicate their higher attractiveness for cTnI detection techniques. The values of kinetic constants should be considered during selection of an optimal clone for a particular type and format of a test system to be developed. One may mention, for example, that 16A11 and 4C2 clones have similar equilibrium constants but different kinetic constants. That indicates that under the same conditions 16A11 clone will form a complex with the antigen faster than 4C2 clone. Yet, it will also dissociate faster if no antigen is available in the microenvironment.

In this view, the clones having the highest values of kinetic constant of association (19C7 and 16A11 clones) are preferable for express one-step assays due to rapid formation of the antibody–antigen bond. At the same time, for constructing ultrasensitive multi-stage systems, which include washing and relatively long incubation steps, lower values of kinetic constant of dissociation are desirable to keep a maximal possible amount of the captured antigen by the moment of its detection at the final stage (19C7 and 4C2 clones).

It is of note that the kinetic parameters of a free antibody may differ upon its sorption on the nanotag surface. Firstly, the magnetic nanotag may carry loads of antibody molecules, and that leads to the nanotag polyvalence [32]. In addition, the nanotag diffusion properties are notably dissimilar to those inherent to molecules. Finally, the non-oriented covalent immobilization onto a spheric surface of the magnetic nanotag may also affect the kinetic properties. Nevertheless, the kinetic properties most commonly correlate [36] so the construction of the most promising magnetic nanotags requires monoclonal antibodies of maximal efficiency. The microarray-based label-free approaches seem to be the most suitable for the high-throughput characterization of a large variety of monoclonal antibodies, while the obtained magnetic nanotags can be comprehensively investigated with, inter alia, label-based techniques.

The developed approach enables other important aspects of antibody characterization, namely, investigation of their specificity and binding with non-target molecules, e.g., heterogeneic entities and other cardiac markers. To demonstrate this option, each one of the tested alternative cardiac markers (heart fatty acids binding protein—hFABP; N-terminal prohormone of brain natriuretic peptide—NT-proBNP) at a concentration of 3500 ng/mL was pumped at the stage of antigen passing. No one of the six studied antibodies exhibited a statistically reliable increase in biolayer thickness, which would indicate binding with hFABP or NT-proBNP. Apparently, that was due to the fact that the studied commercial antibodies were carefully investigated by the recognized manufacturer and selected for high specificity (among other properties).

### 2.4. Characterization of Tracer Antibody. Selection of the Optimal Pair of Capture/Tracer Antibodies

One of the most widely used and efficient formats realized in developments of highly sensitive test systems for cTnI detection is the sandwich immunoassay. In this format, a capture antibody, which specifically binds to one cTnI epitope, is immobilized onto a solid phase (including mobile solid phases such as micro- and nanoparticles). A tracer antibody, which enables forming the signal to be registered, is an antibody to another cTnI epitope. It should be noted that inadequate selection of the tracer antibody may prevent a proper assembly of the “capture antibody–antigen–tracer antibody” complex.

The next experiments were devoted to kinetic characterization of the tracer antibody that recognized the cTnI antigen comprised within its complex with the capture antibody selected at the previous stage. For this purpose, the same complexes of “immobilized capture antibody–antigen” were formed in each registration spot of the mSCI sensor chip. The flow cell of the mSCI biosensor was replaced with one that had multiple independent fluidic channels. Then, different tracer antibodies were simultaneously pumped along the registration spots through these fluidic channels. Five different clones (16A11, 4C2, M155, MF4, and 8E10) of the tracer antibodies were tested paired with the capture antibody of 19C7 clone (Table 2). The kinetic constants for interaction of the tracer antibodies with the immune complex were calculated as described above for characterization of the capture antibodies.

A comparison of the results in Table 1 and Table 2 indicates that the same clones of the same antibody may exhibit substantially different values of kinetic constants depending on capture/tracer function and, correspondingly, interaction with the antigen in its free form or comprised in an immune complex containing another clone of antibody. The difference may be due to dissociation of the immune complex during investigation of the tracer antibody interactions. Therefore, the tracer antibody constants shown in Table 2 are influenced by the kinetic parameters of the capture antibody (clone 19C7), primarily, its kinetic constant of dissociation.

Another significant affecting factor is mutual spatial arrangement of the epitopes for capture and tracer antibodies. The antibodies can recognize the spatially adjacent and even partially overlapping epitopes. Hence, on one hand, the capture antibody already bound with the antigen may hinder the tracer antibody binding. On the other hand, hypothetically, the binding of tracer antibody may displace the capture antibody. As can be seen from Table 1, M155 clone itself has a relatively high equilibrium constant of dissociation (K_D_ = 2.96 × 10^−8^ M). However, the same clone shows considerably smaller parameters of binding (K_D_ = 1.42 × 10^−6^ M) if used as the tracer antibody in combination with the clone 19C7 employed as the capture antibody. Such dramatic difference may be due to targeting of these clones to adjacent epitopes separated by merely four amino acids: M155 recognizes the fragment of cTnI protein from the 26th to 36th amino acids, while 19C7—from 40th to 50th [55]. Therefore, if the antigen molecules are bound to the antibody clone 19C7 immobilized on the sensor chip surface, the antigenic epitopes for binding M155 antibody clone may be sterically hindered.

Importantly, in view of characterization of tracer antibodies for developments of sandwich immunoassays, the data shown in Table 2, which represent the kinetic parameters of binding with antigen comprised in an immune complex, are more useful than the constants, which describe interactions with the free antigen, because the former take into account the abovementioned possible mismatches between the paired caption and tracer antibodies.

Among the studied candidates for the tracer antibody, 16A11 clone exhibited the best kinetic characteristics as a pair to 19C7 clone as a capture antibody. This pair was used for the next experiments.

### 2.5. Using Selected Antibodies in Development of Rapid Sensitive Assay for cTnI Detection Based on Magnetic Nanotags

To realize a rapid sandwich-type immunoassay for quantitative detection of cTnI, we chose a lateral flow format (Figure 4a,b) based on magnetic nanotags registered by their non-linear remagnetization with a highly sensitive technique based on magnetic particle quantification (MPQ) principle (see details in the Materials and Methods section). The capture antibodies of 19C7 clone were covalently immobilized on a surface of 200-nm commercially available superparamagnetic particles using a carbodiimide method [56,57,58]. The particles were added to the analyzed sample, where, if cTnI was present, the “19C7 clone antibody—Antigen” immune complexes formed. Then, a test strip with the pre-deposited antibody of 16A11 clone was immersed into the sample. That antibody bound to the antigen within the immune complex together with the magnetic nanotags. After the assay, the test strip was placed into the MPQ reader to measure the quantity of superparamagnetic particles bound at the test line, where the tracer antibody was deposited. The larger cTnI content in the sample, the higher the recorded magnetic signal.

The assay conditions were optimized to maximize the ratio of specific magnetic signal in the antigen presence to non-specific magnetic signal under the antigen absence. The following parameters were optimized (see details in Appendix A): amount of capture antibody immobilized on the magnetic particles, amount of magnetic particles added to the analyzed sample, concentration of tracer antibody deposited on the test strip. The optimized parameters are given in Section 3.5.

Figure 4c exhibits a calibration plot as a dependence of magnetic signal from the lateral flow test strip upon concentration of cTnI spiked into commercial samples of troponin-free human serum. The limit of detection determined using this plot is 0.08 ng/mL. The dynamic range exceeds three orders of concentration magnitude. The demonstrated analytical parameters are not inferior to those of the traditional techniques of cTnI detection, including labor- and time-consuming ELISA-based approaches [59,60] and a typical lateral flow assay based on gold labels, which detects cTnI at the level of 1 ng/mL or worse [61]. In addition, the developed immunoassay is rapid (25 min) and easy-to-use because of the immunochromatographic principle. The assay specificity was confirmed by the absence of false-negative signals during analysis of serum samples that contained, instead of cTnI, the following non-target molecules, including other cardiomarkers (see details in Appendix A): fatty-acid-binding cardiac protein (1000 ng/mL), pro-brain natriuretic peptide (1000 ng/mL), thyrotropic hormone (300 ME/mL), biotin (3000 ng/mL), and chloramphenicol (100 ng/mL).

## 3. Materials and Methods

### 3.1. Reagents

The following reagents were used in the experiments: monoclonal antibodies (M155, 16A11, MF4, 19C7, 8E10, and 4C2 clones) against cTnI; cTnI antigen; cTnI-free pooled anonymized human serum (HyTest Ltd., Turku, Finland and Moscow, Russia); Unisart^®^ CN-140 nitrocellulose lateral flow membrane (Sartorius AG, Goettingen, Germany); absorbent pad (MDI, Ambala Cantt, India); *N*-hydroxysulfosuccinimide sodium salt (sulfo-NHS), 3-aminopropyltriethoxy- silane (APTES), 3-Glycidyloxypropyltrimethoxysilane (GLYMO), bovine serum albumin (BSA), *N*-ethylcarbodiimide hydrochloride (EDC) and 2-(N-morpholino)ethanesulfonic acid (MES) (Sigma-Aldrich, Taufkirchen, Germany); Triton X-100 (Panreac, Barcelona, Spain); dimethylformamide (DMF), sodium chloride, potassium dihydrophosphate, isopropyl alcohol, methanol, potassium hydroxide and hydrochloric acid and hydrogen peroxide (Chimmed, Moscow, Russia); sulfuric acid (Sigma-Tech, Balashikha, Russia). Superparamagnetic carboxyl-modified (COOH-) particles of 200 nm in diameter (“Bio-Estapor Microspheres”) were purchased from Estapor—Merck Millipore, Fontenay sous Boi, France. All other chemicals were of analytical grade.

### 3.2. Cleaning of Microscope Cover Glass Slip Surface

Immediately prior to chemical modifications, microscope cover glass slips were washed with methanol and then treated with a mixture of 30% hydrogen peroxide and 95% sulfuric acid (1:3) for 40 min at 40 °C [62]. After cooling, they were successively washed three times with tridistilled water, twice with acetone, and once with methanol.

### 3.3. Epoxylation of the Glass Surface

After cleaning, the glass surface was washed three times with tridistilled water and two times with methanol followed by incubation in a 5% GLYMO solution in methyl alcohol for 16 h under hood. After the incubation, the glass slip was heated in a drying oven at 105 °C for 60 min followed by washing with dimethyl sulfoxide and methyl alcohol. The modified glass slips were stored at room temperature.

### 3.4. Procedure of Kinetic Characterization of Antibodies

#### 3.4.1. Capture Antibodies

Different monoclonal antibodies were deposited as 0.1 μL microdots (1 mg/mL in PBS, pH 7.4) onto different registration spots of the epoxylated sensor chip. Only one of the antibody clones M155, 16A11, MF4, 19C7, 8E10, and 4C2 was deposited in each registration spot. Then, a solution of 300 ng/mL cTnI (PBS, BSA 0.5%, Tween-20 0.1% pH 7.4) was pumped along the sensor chip surface at a flow rate of 15 μL/min in the flow cell of the mSCI-biosensor. After 30 min, a buffer solution (PBS, BSA 0.5%, Tween-20 0.1% pH 7.4) was pumped at the same flow rate to detect dissociation of the immune complexes formed at the previous step. 

#### 3.4.2. Tracer Antibodies

For characterization of tracer antibodies, an mSCI-biosensor was equipped with a flow cell with several independent fluidic channels. The same clone of monoclonal antibodies (19C7) was deposited as 0.1 μL microdots (1 mg/mL in PBS, pH 7.4) in each registration spot on a single epoxylated sensor chip. Then, the antigen in concentration of 1000 ng/mL was pumped along the sensor chip. After that, the other mAb clones, namely, M155, 16A11, MF4, 8E10, and 4C2 were pumped, each through a separate fluidic channel, to independently characterize the kinetics of antigen-mAb interactions for each clone in different registration spots.

### 3.5. Lateral Flow Test Strips

A 40-mm wide large-pore nitrocellulose membrane with 100-µm clear polyester backing and capillary speed of 140 s/40 mm was used for the test strips. The membrane and absorbent pad were assembled on a backing card. After that, a test line was deposited at a density of 1 µL/cm using the 16A11 antibody clone at concentration of 2 mg/mL. The assembled card was cut into 3-mm wide test strips. The optimal parameters of the lateral flow assay were as follows: the amount of capture antibody immobilized on the magnetic particles—50 µg of the antibody per 1 mg of the particles, the amount of magnetic particles added to the analyzed sample—5 µg, the concentration of tracer antibody deposited on the test strip—2 mg/mL.

### 3.6. Method of Magnetic Particle Quantification

The magnetic nanotags captured at the test line were counted by a detector based on the magnetic particle quantification method [56,58], which is described in detail in [63,64] and realized for this study using modern low-noise electronic components. Briefly, the MPQ method registers a non-linear response of superparamagnetic materials subjected to an alternating magnetic field. The field is generated at two different frequencies, and the registration is carried out at a linear combination of these two frequencies. The MPQ benefits include high-sensitive detection of magnetic nanotags over the whole volume of a sample, even if it is surrounded by various biological materials. The alternating magnetic field in the MPQ detector used in this research comprised a 154-Hz and 150-kHz components of 144 Oe and 56 Oe in amplitudes, respectively. The temporal resolution of the used detector was 1 s, limit of detection—0.4 ng of magnetic material inside the volume of 0.2 mL, detection range—not less than seven orders of magnitude.

### 3.7. Data Processing

Each experiment was performed at least in triplicates. The points on graphs show averaged values, and error bars—standard deviations. For the sensorgram fitting, the least squares method was used. The analytical limit of detection was calculated using the 2σ criterion [65,66] as the cTnI concentration, under which the specific signal becomes twice as large as the standard deviation of the average signal of a sample without cTnI.

## 4. Conclusions

The developed mSCI-biosensor enables multiplex high-throughput characterization of various monoclonal antibodies to assess simultaneously and independently the kinetic and equilibrium constants of their interaction with the antigen. Capture and tracer antibodies were investigated in the multi-channel mode, and the most efficient pairs were selected for different assay types. A mutual competition of the clone pairs for their targeting the adjacent or overlapping epitopes was estimated along with their steric hindrance. The obtained information can be used in developments of rapid sensitive methods of antigen registration, including sandwich immunoassays.

The selected optimal pair of monoclonal antibodies to cTnI was used in the development of a promising express lateral flow method for detection of cardiac troponin I based on functionalized magnetic nanotags and their registration by non-linear remagnetization.

The proposed concept is universal. It allows efficient and rational development of novel immunoanalytical biosensors for a wide range of both low- and high-molecular-weight antigens for in vitro diagnostics (including point-of-care applications and implementation of other techniques for registration of magnetic nanotags [67,68,69,70], e.g., for SARS-CoV-2 sensing [71]), veterinary, environmental monitoring, and food control.

## Figures and Tables

**Figure 1 ijms-23-04474-f001:**
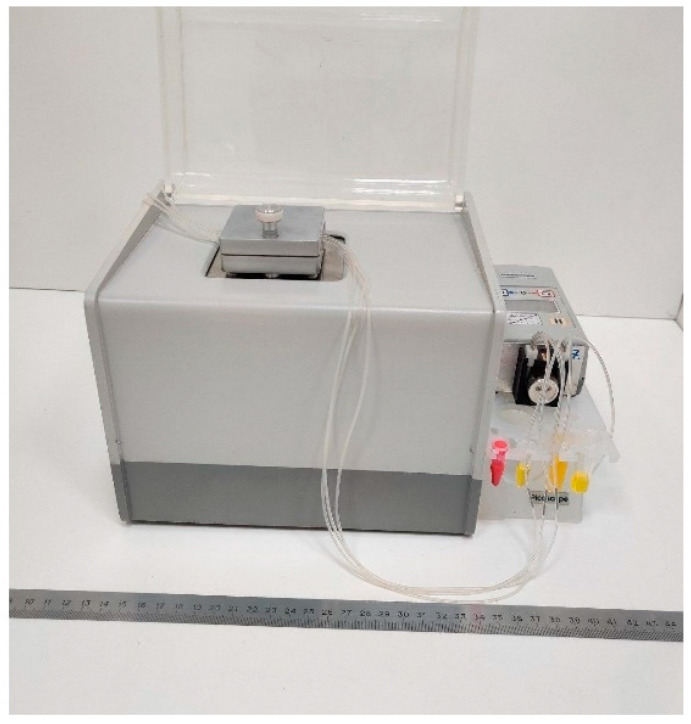
Photograph of the multiplex label-biosensor based on the spectral-correlation interferometry.

**Figure 2 ijms-23-04474-f002:**
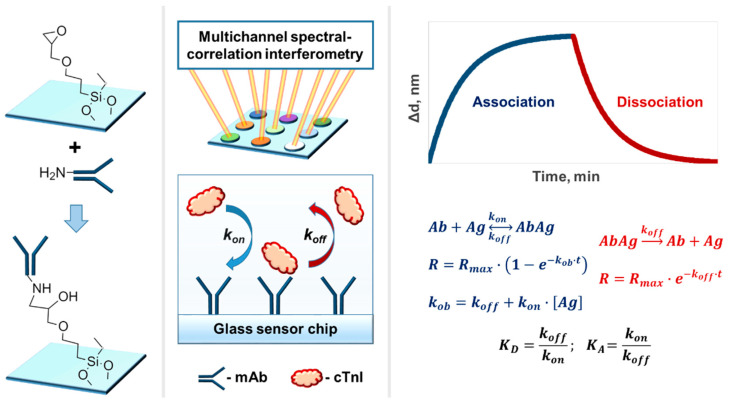
Experimental setup for simultaneous determination of kinetic constants of multiple monoclonal antibodies with an mSCI-biosensor.

**Figure 3 ijms-23-04474-f003:**
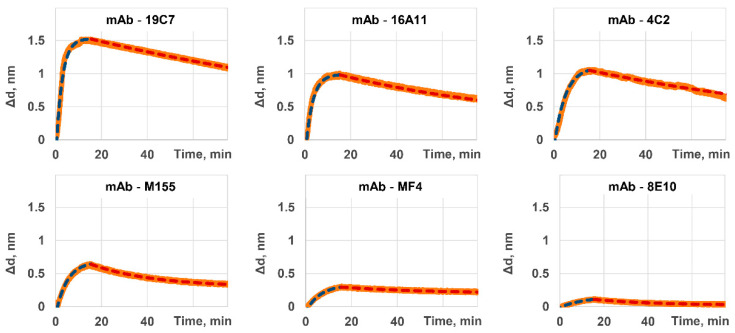
Sensorgrams recorded by the mSCI-biosensor show the stages of association and dissociation of the cTnI antigen with various mAb clones.

**Figure 4 ijms-23-04474-f004:**
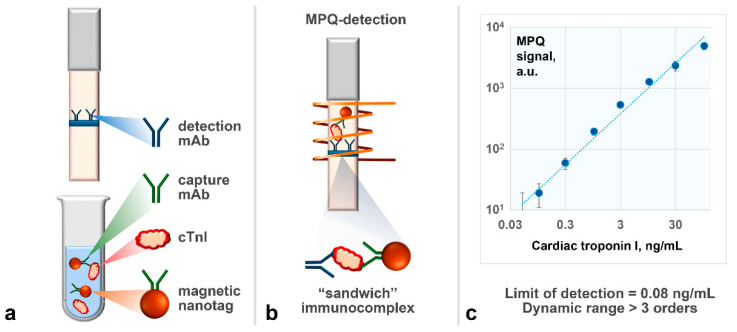
Scheme of the lateral flow sandwich immunoassay based on magnetic nanotags (**a**,**b**) and calibration plot as a dependence of magnetic signal from the lateral flow test strip upon concentration of cTnI spiked into commercial samples of troponin-free human serum (**c**).

**Table 1 ijms-23-04474-t001:** Equilibrium and kinetic constants calculated for different clones of mAb to cTnI used as a capture antibody.

mAb Clone	k_on_ × 10^−5^, M^−1^s^−1^	k_off_ × 10^4^, s^−1^	K_A_ × 10^−7^, M	K_D_ × 10^9^, M^−1^
19C7	11.9 ± 0.8	2.76 ± 0.36	42.9 ± 8.7	2.33 ± 0.79
4C2	6.54 ± 0.72	3.49 ± 0.21	18.8 ± 3.2	5.33 ± 1.39
16A11	10.3 ± 2.1	5.55 ± 0.44	18.5 ± 5.19	5.40 ± 1.46
M155	6.24 ± 0.69	18.5 ± 3.0	3.37 ± 0.91	29.6 ± 8.0
MF4	4.95 ± 0.54	13.0 ± 2.60	3.81 ± 1.18	26.3 ± 6.7
8E10	2.54 ± 0.46	18.6 ± 3.2	1.37 ± 0.48	72.8 ± 25.5

**Table 2 ijms-23-04474-t002:** Kinetic and equilibrium constants calculated for a variety of clones of mAb to cTnI used as a tracer antibody paired up with 19C7 clone served as a capture antibody.

mAb Clone	k_on_ × 10^−5^, M^−1^s^−1^	k_off_ × 10^4^, s^−1^	K_A_ × 10^−7^, M^−^^1^	K_D_ × 10^9^, M^1^
4C2	3.03 ± 0.21	6.99 ± 0.91	4.34 ± 0.87	23.05 ± 4.61
16A11	5.12 ± 0.61	9.22 ± 1.01	5.55 ± 1.28	18.02 ± 4.14
M155	0.17 ± 0.02	24.7 ± 4.7	0.07 ± 0.02	1419 ± 397
MF4	2.48 ± 0.30	19.7 ± 3.7	1.26 ± 0.39	79.5 ± 24.6
8E10	1.67 ± 0.08	23.4 ± 3.5	0.72 ± 0.14	140 ± 28

## Data Availability

Not applicable.

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
