# Peer review of "Multiplex Label-Free Kinetic Characterization of Antibodies for Rapid Sensitive Cardiac Troponin I Detection Based on Functionalized Magnetic Nanotags"

_ijms, 2022, doi:10.3390/ijms23094474_

Round 1
Reviewer 1 Report
The paper by Alexey V. Orlov and colleagues entitled “Multiplex label-free kinetic characterization of antibodies for rapid sensitive cardiac troponin I detection based on functionalized magnetic nanotags” is interesting, original and important. The main content is relatively shower but it is fine for the "communication" style. The manuscript is well organized and well written. Thus, my general recommendation is “accept”. However, I would like to give several comments listed below with the aim to help the authors to improve the manuscript.
- While the sensing method is label-free multiplex spectral correlation interferometry, the test rapid sensitive assay was performed with the lateral flow assay based on magnetic measuring of the magnetic signal of nanotags. It is not clear from the introduction, what is the advantage of this magnetic sensor? In literature is reported on plenty of ultrasensitive magnetic sensors for biomedical applications.
- The authors suggested a label-free characterization method of antibody kinetics, however when antibodies are attached to the magnetic nanotags, this kinetic can change. Did the authors check this point?
- Can authors specify the properties of magnetic nanoparticles? The term “superparamagnetic” appears only in keywords.
- The authors optimized the following parameters of the assay: “amount of capture antibody, amount of magnetic particles, the concentration of tracer antibody deposited on the test strip”, however, optimized parameters are not presented.
- In the Introduction, the authors declared the superior properties of the approach as “assay time – 25 min, limit of cTnI detection in human serum – 0.08 ng/ml, the dynamic range more than 3 orders of concentration magnitude”. Can the authors compare those values with commercially available solutions or those reported earlier in the literature?
Author Response
Dear Reviewer,
First of all, we would like to thank you for the time and efforts for reviewing our manuscript ijms-1673082 "Multiplex label-free kinetic characterization of antibodies for rapid sensitive cardiac troponin I detection based on functionalized magnetic nanotags" by Alexey V. Orlov, Juri A. Malkerov, Denis O. Novichikhin, Sergey L. Znoyko, Petr I. Nikitin.
We have revised the manuscript according to your remarks. The point-by-point description of the improvements is outlined below. Your remarks are underlined; our replies are given as usual text, and the manuscript is cited in italic.
- While the sensing method is label-free multiplex spectral correlation interferometry, the test rapid sensitive assay was performed with the lateral flow assay based on magnetic measuring of the magnetic signal of nanotags. It is not clear from the introduction, what is the advantage of this magnetic sensor? In literature is reported on plenty of ultrasensitive magnetic sensors for biomedical applications.
We have introduced to the Introduction section the following fragment that highlights the advantages of the magnetic sensor used for readout of the lateral flow strips
The MPQ reader compares favorably to other magnetic sensors as readers for LF test strips due to the record-breaking sensitivity of 60 zM [48] and unrivalled 7-order linear range of quantitation of magnetic nanotags irrespective to their color and depth inside the lateral flow membrane [49].
- The authors suggested a label-free characterization method of antibody kinetics, however when antibodies are attached to the magnetic nanotags, this kinetic can change. Did the authors check this point?
The following discussion has been added to section “2.3. Measuring the kinetic and equilibrium constants for capture antibodies”:
It is to note that the kinetic parameters of a free antibody may differ upon its sorption on the nanotag surface. Firstly, the magnetic nanotag may carry loads of antibody molecules, and that leads to the nanotag polyvalence [32]. Besides, the nanotag diffusion properties are notably dissimilar to those inherent to molecules. Finally, the non-oriented covalent immobilization onto a spheric surface of the magnetic nanotag may also affect the kinetic properties. Nevertheless, the kinetic properties most commonly correlate [36] so the construction of the most promising magnetic nanotags requires monoclonal antibodies of maximal efficiency. The microarray-based label-free approaches seem to be the most suitable for the high-throughput characterization of a large variety of monoclonal antibodies, while the obtained magnetic nanotags can be comprehensively investigated with, inter alia, label-based techniques.
- Can authors specify the properties of magnetic nanoparticles? The term “superparamagnetic” appears only in keywords.
We have introduced to the Section 3.1 the following information about the magnetic nanoparticles and their manufacturer:
Superparamagnetic carboxyl-modified (COOH-) particles of 200 nm in diameter (“Bio-Estapor Microspheres”) were purchased from Estapor—Merck Millipore, France.
Additional information about the properties of these commercially available nanoparticles can be obtained from the manufacturer. We have also emphasized the term "superparamagnetic" in several fragments of the main text.
- The authors optimized the following parameters of the assay: “amount of capture antibody, amount of magnetic particles, the concentration of tracer antibody deposited on the test strip”, however, optimized parameters are not presented.
Firstly, we have added the following sentence to the mentioned fragment:
The optimized parameters are given in Section 3.5.
The requested optimized values have been added to Section 3.5:
The optimal parameters of the lateral flow assay were as follows: the amount of capture antibody immobilized on the magnetic particles – 50 µg of the antibody per 1 mg of the particles, the amount of magnetic particles added to the analyzed sample – 5 µg, the concentration of tracer antibody deposited on the test strip – 2 mg/mL.
Besides, we have added as a Supplementary Materials the experimental data that support the optimization process along with proper referencing in the main text.
- In the Introduction, the authors declared the superior properties of the approach as “assay time – 25 min, limit of cTnI detection in human serum – 0.08 ng/ml, the dynamic range more than 3 orders of concentration magnitude”. Can the authors compare those values with commercially available solutions or those reported earlier in the literature?
Beside the previously available comparison with the ELISA-based approaches and references [59] and [60], we have added a comparison with commercial approaches as follows:
The demonstrated analytical parameters are not inferior to those of the traditional techniques of cTnI detection, including labor- and time-consuming ELISA-based approaches [59,60] and a typical lateral flow assay based on gold labels, which detects cTnI at the level of 1 ng/mL or worse [61].
Thank you.
On behalf of the authors,
corresponding author of this manuscript
Dr. Petr Nikitin
Head of Biophotonics Laboratory
General Physics Institute
Russian Academy of Sciences
Reviewer 2 Report
The authors developed mSCI-biosensor that enables multiplex high-throughput characterization of various monoclonal antibodies for characterization. With the selected optimal pair of monoclonal antibodies to cTnI, lateral flow strip sensor was developed for detection of cardiac troponin I. The sensor strip was based on functionalized magnetic nanotags and its limit of detection was demonstrated to be 0.08 ng/ml.
The manuscript can be re-considered for publication in the International Journal of Molecular Sciences after revision with comments below.
- What makes the “multiplex spectral correlation interferometry” measure the many different targets simultaneously? The principle of the interferometry and the reason why it can be used for the multiplex should be described.
- For the characterization of the antibodies, the tests for the specificity of the antibodies with other cardiac markers should be added.
- In Figure 2, results from different antibodies can be merged for clearer comparison instead of arranging the datas with different y-axis.
- How could the density and concentration be decided for making the sensor strip? Even though It is mentioned that the assay conditions were optimized to maximize the ratio of specific magnetic signals, there is not any supporting data for the optimization. The optimization process should be supported by data.
- The limit of detection of the strip sensor with magnetic particles is 0.08 ng/ml. What makes the MPQ be more sensitive than others with 3 orders of high limit of detection ? The principle of MPQ is missing even in the Materials and Methods section.
- The information of the MPQ reader should be added for more clarification.
- It is mentioned that the assay for specificity was tested but the data with the other cardiac markers is not presented. The data related to the specificity should be added.
Author Response
Dear Reviewer,
First of all, we would like to thank you for the time and efforts for reviewing our manuscript ijms-1673082 "Multiplex label-free kinetic characterization of antibodies for rapid sensitive cardiac troponin I detection based on functionalized magnetic nanotags" by Alexey V. Orlov, Juri A. Malkerov, Denis O. Novichikhin, Sergey L. Znoyko, Petr I. Nikitin.
We have revised the manuscript according to your remarks. The point-by-point description of the improvements is outlined below. Your remarks are underlined; our replies are given as usual text, and the manuscript is cited in italic.
- What makes the “multiplex spectral correlation interferometry” measure the many different targets simultaneously? The principle of the interferometry and the reason why it can be used for the multiplex should be described.
For better perception, we have renamed the previous Section 3.4 Biosensor based on the multiplex spectral correlation interferometry, which contained a detailed description of the multiplex spectral correlation interferometry and respective references, and moved it to the beginning of the Results and Discussion section (now this is Section 2.1 Multiplex spectral correlation interferometry).
- For the characterization of the antibodies, the tests for the specificity of the antibodies with other cardiac markers should be added.
To address this remark, we have added to Section 2.3 the following fragment:
The developed approach enables other important aspects of antibody characterization, namely, investigation of their specificity and binding with non-target molecules, e.g., heterogeneic entities and other cardiac markers. To demonstrate this option, each one of the tested alternative cardiac markers (heart fatty acids binding protein – hFABP; N-terminal prohormone of brain natriuretic peptide – NT-proBNP) at concentration of 3500 ng/mL was pumped at the stage of antigen passing. No one of the six studied antibodies exhibited a statistically reliable increase in the biolayer thickness, which would indicate binding with hFABP or NT-proBNP. Apparently, that was due to the fact that the studied commercial antibodies were carefully investigated by the recognized manufacturer and selected for high specificity (among other properties).
- In Figure 2, results from different antibodies can be merged for clearer comparison instead of arranging the datas with different y-axis.
We have revised the mentioned figure (actual figure number is #3) to have the same Y-axes for easier comparison. The authors tried the suggested merging of the 6 plots but found that it substantially complicates the data perception and comparison due to overlapping the curves exhibiting similar kinetics.
- How could the density and concentration be decided for making the sensor strip? Even though It is mentioned that the assay conditions were optimized to maximize the ratio of specific magnetic signals, there is not any supporting data for the optimization. The optimization process should be supported by data.
We have added the data that support the optimization process as Supplementary Materials with proper referencing in Section 2.5 of the main text.
- The limit of detection of the strip sensor with magnetic particles is 0.08 ng/ml. What makes the MPQ be more sensitive than others with 3 orders of high limit of detection ? The principle of MPQ is missing even in the Materials and Methods section.
We have revised the previous Section 3.7 “Method of magnetic particle quantification (MPQ)” (now section 3.6) to add more details about the MPQ method as well as the references on papers devoted to this registration technique as follows:
The magnetic nanotags captured at the test line were counted by a detector based on the magnetic particle quantification method [56,58], which was described in details in [63,64]. Briefly, the MPQ method registers a non-linear response of superparamagnetic materials subjected to an alternating magnetic field. The field is generated at two different frequencies, and the registration is done at a linear combination of these two frequencies. The MPQ benefits include high-sensitive detection of magnetic nanotags over the whole volume of a sample, even it is surrounded by various biological materials. The alternating magnetic field in the MPQ detector used in this research comprised a 154-Hz and 150-kHz components of 144 Oe and 56 Oe in amplitudes, respectively. The temporal resolution of the used detector was 1 s, limit of detection – 0.4 ng of magnetic material inside the volume of 0.2 mL, detection range – not less than 7 orders of magnitude.
Besides, a brief fragment about the advantages of the MPQ reader has been added to the Introduction section (please see the reply to your next remark).
- The information of the MPQ reader should be added for more clarification.
Along with the information on the MPQ principle, which has been added to the Materials and Methods section according to your remark #5 (see above), we have added the following fragment and related references to the Introduction:
The MPQ reader compares favorably to other magnetic sensors as readers for LF test strips due to the record-breaking sensitivity of 60 zM [48] and unrivalled 7-order linear range of quantitation of magnetic nanotags irrespective to their color and depth inside the lateral flow membrane [49].
- It is mentioned that the assay for specificity was tested but the data with the other cardiac markers is not presented. The data related to the specificity should be added.
We have added the data on the specificity testing (including the non-target cardiac markers hFABP and NT-proBNP) to Supplementary Materials along with the reference to that in the main text.
Thank you.
On behalf of the authors,
corresponding author of this manuscript
Dr. Petr Nikitin
Head of Biophotonics Laboratory
General Physics Institute
Russian Academy of Sciences
Round 2
Reviewer 1 Report
I am satisfied with the authors' answers and recommend the article for publication in its present form.
Reviewer 2 Report
The manuscript is revised well, it can be published in International Journal of Molecular Sciences.